# Multimodal Image-Text Matching Improves Retrieval-based Chest X-Ray Report Generation

**Jaehwan Jeong**[*1]                                   JAEHWANJ@STANFORD.EDU

**Katherine Tian**[*2]                                 KTIAN@COLLEGE.HARVARD.EDU

**Andrew Li**[1]                                         ANDREWLI@STANFORD.EDU

**Sina Hartung**[3]                                 SINAHARTUNG@HMS.HARVARD.EDU

**Fardad Behzadi**[4]                              FBEHZADI1@BWH.HARVARD.EDU

**Juan Calle**[5]                                        CALLETORO@UTHSCSA.EDU

**David Osayande**[4]                               DOSAYANDE@BWH.HARVARD.EDU

**Michael Pohlen**[1]                                      POHLEN@STANFORD.EDU

**Subathra Adithan**[6]                           SUBATHRA26@GMAIL.COM

**Pranav Rajpurkar**[3]                     PRANAV_RAJPURKAR@HMS.HARVARD.EDU

[1] *Stanford University*

[2] *Harvard University*

[3] *Harvard Medical School*

[4] *Brigham and Women's Hospital*

[5] *University of Texas Health Science Center at San Antonio*

[6] *Jawaharlal Institute of Postgraduate Medical Education and Research*

**Editors:** Accepted for publication at MIDL 2023

## Abstract

Automated generation of clinically accurate radiology reports can improve patient care. Previous report generation methods that rely on image captioning models often generate incoherent and incorrect text due to their lack of relevant domain knowledge, while retrieval-based attempts frequently retrieve reports that are irrelevant to the input image. In this work, we propose Contrastive X-Ray REport Match (X-REM), a novel retrieval-based radiology report generation module that uses an image-text matching score to measure the similarity of a chest X-ray image and radiology report for report retrieval. We observe that computing the image-text matching score with a language-image model can effectively capture the fine-grained interaction between image and text that is often lost when using cosine similarity. X-REM outperforms multiple prior radiology report generation modules in terms of both natural language and clinical metrics. Human evaluation of the generated reports suggests that X-REM increased the number of zero-error reports and decreased the average error severity compared to the baseline retrieval approach. Our code is available at: https://github.com/rajpurkarlab/X-REM

**Keywords:** Medical Report Generation, Vision-Language Modeling, Clinical Expert Evaluation

---

[*] Contributed equally

## 1. Introduction

The ability to generate radiology reports automatically using machine learning models can dramatically improve clinical workflows and patient care (Boag et al., 2020; Hartung et al., 2020). Such models can triage cases, lessen radiologist workloads, and reduce delays in diagnosis (Nsengiyumva et al., 2021; Dyer et al., 2021; Annarumma et al., 2019). Additionally, AI tools, whether acting autonomously or collaborating with radiologists, can improve human accuracy (Seah et al., 2021; Sim et al., 2020).

However, current report generation models are yet to match expert-level performance and are not ready for deployment in clinical practice (Jing et al., 2018; Chen et al., 2020). While many prior methods adopt image-captioning models to generate reports from an image input, the generated texts often contain hallucinated information and self-contradictory claims (Ramesh et al., 2022). Natural language generative models lack the medical knowledge necessary to self-evaluate the sanity of the generated reports and are therefore prone to hallucination. Another stream of work approaches medical report generation as an image-text retrieval task, which retrieves a report that best describes the input image from a medical corpus of radiology reports. The biggest advantage of the retrieval method is that the clinical coherency of human-written reports is guaranteed (Endo et al., 2021). Since the variety of medical diagnoses present in radiology reports is bounded, a large retrieval corpus can provide sufficient coverage of potential diagnoses of an input X-ray image. Yu et al. (2022) has shown that while the theoretical upper bound for the performance of a retrieval-based generation method is high, prior approaches are still far from the upper bound as they often fail to select the ideal report due to the coarse similarity metric used during the retrieval step. Our work aims to narrow this performance gap.

We propose Contrastive X-Ray REport Match (X-REM), an innovative, retrieval-based radiology report generation method. In addition to the aforementioned benefits of the retrieval-based approach, X-REM differs from the current paradigm in two key ways. First, our approach leverages a language-image model to acquire multimodal representations of image and text, as opposed to representing them separately using two unimodal encoders. Secondly, we use a novel image-text matching score as the main similarity metric for the report retrieval. In addition to aligning the image and text embeddings in the pre-training phase, we also perform supervised contrastive learning that fine-tunes the model to match image and text with the same clinical label. As image-text matching score is a learned similarity metric tuned to match X-ray images and radiology reports based on their medical features, it can better capture complicated multi-modal interactions than embedding cosine similarity, the metric conventionally adopted by previous retrieval-based works.

When evaluated on the pre-processed test split of MIMIC-CXR (Johnson et al., 2019), X-REM shows a noticeable improvement over previous radiology report generation modules on both clinical and natural language metrics. Additionally, we collected expert annotations of error scores on reports generated by X-REM and the baseline retrieval-based approach. The human evaluation study demonstrates that our model makes significant improvements upon the baseline model.

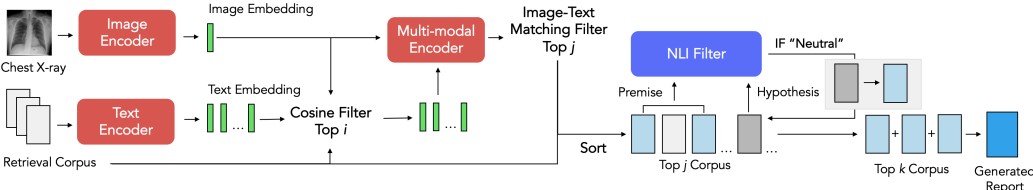

Figure 1: An overview of the inference step of X-REM

## 2. Related Works

Earlier works in radiology report generation have focused heavily on performing image-captioning with a transformer architecture. R2Gen (Chen et al., 2020) and $\mathcal{M}^2$Trans (Miura et al., 2021) use a memory-driven Transformer and a Meshed-Memory Transformer (Cornia et al., 2019), respectively, to directly generate a radiology report. WCL (Yan et al., 2021) shows that incorporating a weakly supervised contrastive loss while training the encoder-decoder models improves the model performance. CvT2DistilGPT2 (Nicolson et al., 2022) demonstrates that pre-trained checkpoints for traditional computer vision and natural language tasks can be helpful for radiology report generation as well. However, one main shortcoming of the image-captioning models is the presence of medically inconsistent information frequently found in the generated reports. To address this issue, CXR-RePaiR (Endo et al., 2021) adopts an image-text retrieval method that retrieves a report whose CLIP (Radford et al., 2021) text embedding scores the highest cosine similarity with the chest X-ray's CLIP image embedding. While CXR-RePaiR uses two pre-trained unimodal encoders to compute the similarity score, X-REM additionally employs a multimodal encoder that has gone through a supervised contrastive learning that matches chest X-ray image and radiology reports based on their clinical labels.

## 3. Method

As shown in Figure 1, X-REM first embeds an input chest X-ray image using the language-image model's image encoder. Next, it filters the retrieval corpus to the top $i$ reports with the highest cosine similarity with the input image. Then, X-REM selects the top $j$ reports with the highest image-text matching score with the input image. X-REM traverses the $j$ reports in the order of decreasing image-text matching score and selects at most $k$ reports, using a natural language inference (NLI) score to filter out reports with repetitive information. Finally, the model concatenates the retrieved reports into a single document delimited by space character.

**Language Image Model**   In this study, we choose ALBEF (Li et al., 2021) as the backbone language-image model for X-REM. ALBEF has three encoders: one image and one text encoder that each generates an embedding of the input image and text, and one multimodal encoder that fuses the output of the two preceding encoders to produce the image-text matching scores. As ALBEF directly aligns the embeddings of its image and text encoders, we can use its unimodal encoders to directly compute the cosine similarity of the input image and text as well. For our study, we use the default ALBEF architecture that consists of a BERT$_{base}$ (Devlin et al., 2019) with 123.7M parameters and a ViT-B/16 (Dosovitskiy et al., 2020) with 85.8M parameters.

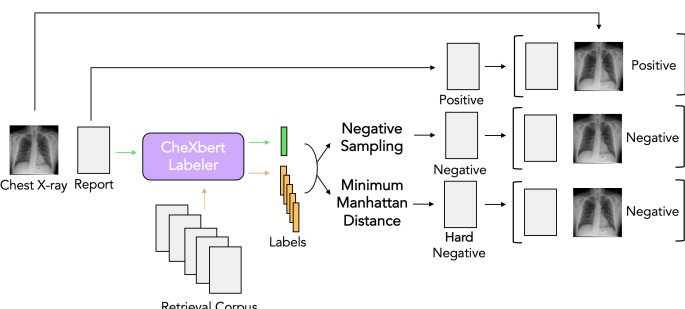

Figure 2: Dataset generation for Image-Text Matching Task

**Image-Text Matching**  Image-text matching is a binary classification task that predicts whether an image and text describe the same event—in our case, the same report. Since image-text matching score can be fine-tuned on a domain-specific dataset, it can use the learned features to make a sophisticated judgment on the inputs' multimodal interaction. In this study, we propose using the logit value for the matching task's `positive` class as the similarity score of the pair. Given a multimodal encoder fine-tuned on image-text matching task $M_{ITM}$, input chest X-ray image $x$, and radiology report $r$, we define the image-text matching score function as

$$f_{ITM}(x, r) = M_{ITM}(x, r)[\texttt{positive}]. \tag{1}$$

As conventional medical datasets naturally consist of positive image-text pairs, synthetic generation of negative samples is necessary for image-text matching learning. Figure 2 shows the process of the dataset generation. After generating a 14-dimensional clinical label of the reports in the pre-training dataset using CheXbert labeler (Smit et al., 2020), we generate two types of negative samples: a sample that randomly matches images and reports with different CheXbert labels and a sample that matches images and reports with the smallest nonzero Manhattan CheXbert label distance.

**Cosine Similarity Filter**  As the computation for image-text matching score may be a resource-heavy process, we take advantage of ALBEF's already-aligned unimodal embeddings and first narrow down the pool of candidate reports to top $i$ number of reports that score high cosine similarity with the input image. Then we compute the image-text matching scores for the $i$ reports to select top $j$ reports with the highest image-text matching scores.

**Natural Language Inference Filter**  We prevent the model from retrieving multiple reports with either overlapping or conflicting content by appending a filter that rejects a candidate report if it is either entailed or contradicted by the generated report. When we have either a contradiction or a redundancy, we prioritize the report that earns higher image-text matching score. Formally, let $k$ represent the desired number of reports retrieved and $B = \{r_1, \ldots, r_m\}$ be the ordered batch of $m$ reports we have selected so far where $f_{ITM}(x, r_a) > f_{ITM}(x, r_b)$ for $a < b$. For a given premise $p$ and hypothesis $h$, let $g(p, h)$ be a language model that classifies the relationship between $p$ and $h$ into one of $\{\texttt{contradiction, entailment, neutral}\}$. Then, for $n > m$, we add $r_n \in B_{ITM}$ to $B$ if

$$m < k \wedge g(B, r_n) = \texttt{neutral}. \tag{2}$$

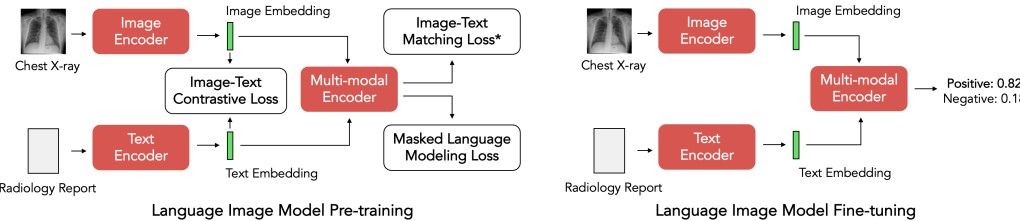

Figure 3: An overview of the training step of the language-image model. Image-Text Matching Loss* in the pre-training step is different from the fine-tuning objective in terms of the negative samples used.

**Learning Objectives**   X-REM is a three-layered module that uses unimodal image and text encoders for measuring cosine similarity, a multimodal encoder for measuring image-text matching score, and a language model for classifying NLI relationship. Figure 3 shows the training process of the encoders. For training the two unimodal encoders, we follow the three pre-training objectives proposed by Li et al. (2021): $\mathcal{L}_{\mathrm{itc}}$, an image-text contrastive loss that directly aligns the unimodal representations of the image and text, $\mathcal{L}_{\mathrm{mlm}}$, a masked language modeling loss that predicts the randomly-masked tokens, and $\mathcal{L}_{\mathrm{itm}}$, an image-text matching loss that predicts whether the given image and text are positively matched. The main difference between the image-text matching loss in the pre-training and the fine-tuning step is the usage of supervised clinical labels for generating the negative samples. We fine-tune both the multi-modal encoder and the language model on image-text matching and natural language inference, respectively, to perform a $n$-way classification with a cross-entropy loss.

**Data**   MIMIC-CXR (Johnson et al., 2019) is a de-identified, publicly available dataset of chest radiographs and semi-structured radiology reports that contains over 225,000 studies from over 65,000 patients. We follow the pre-processing steps suggested by Endo et al. (2021) to combine the train and validation splits into a bigger train split and then extract the impression and findings sections of the radiology reports. Additionally, we reserve 60 studies from the impression section's training set for human evaluation. Our training data for impression, findings, and impression + finding each has 185,538 studies, 123,839 studies, and 123,814 studies with 313,777, 230,436, and 230,392 pairs of chest X-ray images and radiology reports, respectively. The same training sets serve as the retrieval corpus in the inference step. For our test set, we refer to Yu et al. (2022) and remove multiple samples that are from the same study. While filtering out samples from a duplicate study, we prioritize samples with images in either PA or AP view, randomly choosing between them if both views are present. Our final test set for impression, findings, and impression + findings each consists of 2,192, 1,597, and 2,192 pairs of chest X-ray images and radiology reports.

## 4. Experiments

**Implementation**   After loading the published pre-trained weights (ALBEF 4M), we further pre-train ALBEF on MIMIC-CXR for 60 epochs with a learning rate of $1 \times 10^{-5}$. Then, we fine-tune the language-image model on the image-text matching task for 8 epochs

|  | Data | RadCliQ ↓ | RadGraph $F_1$ ↑ | CheXbert ↑ | BERTScore ↑ | BLEU2 ↑ |
|---|---|---|---|---|---|---|
| **X-REM** | I | **3.781** | **0.133** | **0.384** | **0.287** | **0.084** |
| CXR-RePaiR* | I | 4.121 | 0.090 | 0.379 | 0.193 | 0.055 |
| BLIP | I | 4.313 | 0.046 | 0.309 | 0.190 | 0.030 |
| **X-REM** | I + F | **3.835** | **0.172** | **0.351** | **0.287** | **0.161** |
| WCL* | I + F | 3.986 | 0.143 | 0.309 | 0.275 | 0.144 |
| R2Gen* | I + F | 4.051 | 0.134 | 0.286 | 0.271 | 0.137 |
| $\mathcal{M}^2$ Trans†* | F | **3.277** | **0.244** | **0.452** | **0.386** | **0.220** |
| **X-REM** | F | 3.585 | 0.181 | 0.381 | 0.353 | 0.186 |
| CvT2DistilGPT2* | F | 3.617 | 0.183 | 0.375 | 0.347 | 0.196 |

Table 1: Comparison of X-REM to previous report generation models. Models trained on data $I$, $F$, $I + F$ generate the impressions, findings, and both the impression and the findings sections of MIMIC-CXR, respectively ($\mathcal{M}^2$ Trans† has been additionally trained on CheXpert). Results with * are taken from Yu et al. (2022) who evaluated the models on the identically-preprocessed MIMIC-CXR test set.

with the default learning rate of $2 \times 10^{-5}$. We resize the input chest X-ray images into the shape of $(256, 256, 3)$ for pre-training and $(384, 384, 3)$ for fine-tuning and disable the original language-image model's on-the-fly data augmentation steps such as random cropping and flipping. It took us 2 days to pre-train the language-image model and 4 days to fine-tune it on 4 NVIDIA RTX 8000 GPUs. For the language model, we use BERT$_{base}$ with the published checkpoint fine-tuned on MedNLI (Romanov and Shivade, 2018) and RadNLI(Miura et al., 2021).

**Evaluation Metrics** We refer to Yu et al. (2022) and use RadCliQ (Yu et al., 2022), RadGraph $F_1$ (Yu et al., 2022), CheXbert vector similarity (Smit et al., 2020), BERTScore (Zhang et al., 2019), and BLEU (Papineni et al., 2002) as the main metrics for evaluation. The two natural language metrics we use are BLEU-2 and BERTScore. BLEU-2 measures the count of overlapping unigrams and bigrams across two texts. BERTScore metric uses `distillroberta-base` as the base BERT model to compare the token embeddings of the ground truth and input text. For clinical accuracy, we use Radgraph $F_1$ and CheXbert vector similarity. RadGraph $F_1$ metric converts two input radiology reports into knowledge graphs and measures the overlap between the two graphs. CheXbert vector similarity uses a 14-dimensional vector that indicates the presence of 13 common symptoms and the no-finding observation for each report, then computes the cosine similarity between the two vectors. RadCliQ combines BLEU-2 and RadGraph $F_1$. As the RadCliQ score is an estimation of the error, smaller RadCliQ values are preferable. Yu et al. (2022) has recently shown that RadCliQ is the most aligned with the radiologists' evaluation of the reports, so we optimize the model on RadCliQ.

**Comparison** In Table 1, we compare the performance of X-REM with multiple prior radiology report generation modules CXR-RePaiR, WCL, R2GEN, $\mathcal{M}^2$ Trans, and CvT2Distil GPT2 on the preprocessed test split of MIMIC-CXR. As the models were originally trained to generate different sections of a radiology report, we trained X-REM on impression, findings, and findings + impression to make a direct comparison with each of them. Moreover, in order to compare the retrieval approach with the image-captioning method, we generate baseline image-captioning results with BLIP (Li et al., 2022), an image-captioning model

that was pre-trained and fine-tuned on MIMIC-CXR for 50 epochs and 30 epochs respectively. The results indicate that X-REM shows a notable improvement from the previous models measured on BERTScore, CheXbert vector similarity, RadGraph $F_1$, and RadCliQ for generating both impressions and findings + impressions sections of a radiology report. The wide gap between the scores for CXR-RePaiR and X-REM suggests that the image-text matching score serves as a better similarity metric than cosine similarity. The poor performance of BLIP on the metrics demonstrates the limitations of an image-captioning model without the additional learning objectives introduced in the prior radiology report generation modules. For generating findings, while $\mathcal{M}^2$ Trans shows better performance than X-REM, it should be noted that the image encoders of $\mathcal{M}^2$ Trans has been additionally trained on CheXpert (Irvin et al., 2019), making a direct comparison between X-REM and $\mathcal{M}^2$ Trans unfair.

**Expert Evaluation** We conduct a human evaluation on a subset of the impression sections of the 60 studies from MIMIC-CXR and recruit four radiologists to provide evaluations. We note that our study has been approved by an institutional review board. We use CXR-RePaiR as the machine learning baseline as CXR-RePaiR also adopts a retrieval approach to generate the impression sections of reports. For each study image, we compile three reports: one generated from our method (X-REM), one generated from CXR-RePaiR trained on the same MIMIC-CXR impression training set, and one taken from a human benchmark (MIMIC-CXR). To each radiologist, we present one image and one report for each of the 60 studies. Each of these 60 reports is randomly and independently chosen from one of the three sources with probabilities 0.5 X-REM, 0.25 baseline, 0.25 human benchmark. Across all annotators, we had 118 X-REM annotations, 69 baseline annotations, and 53 human benchmark annotations for a total of 240 annotations. The radiologist is blinded to the source, and we ask each radiologist to assess the error severity of their assigned reports.

**Error Scoring:** Each report is broken down into lines. The radiologist is asked to score the error of each line of the report. The radiologists select from five possible error categories (No error, Not actionable, Actionable nonurgent error, Urgent error, or Emergent error), which we map to error scores 0, 1, 2, 3, 4, respectively, in order of severity. We collapse severity per line to a single severity score per annotation using the following metrics: (1) maximum error severity (MES) across all lines in the report, which measures the worst error in the report, and (2) average error severity (AES) per line in the original report. AES is the sum of error severity across lines standardized by the number of lines, which avoids punishing longer reports. Our analysis finds that X-REM increases the number of zero-error reports by 70% and provides statistically strong improvements over the baseline. We present the distribution of MES and AES across annotations for each source in Table 2. According to the radiologists, 18% of reports generated by X-REM have zero mistakes, which surpasses 9% from the baseline but lags behind the human benchmark's 34%.

**Paired Comparison:** In order to directly assess the improvement of X-REM over the baseline, we analyze the 40 studies that have at least one annotation from both X-REM and the baseline.We find that in 62.5% of studies, X-REM has the same or lower MES compared to the baseline. On average, our method reduces MES to 2.11 from 2.29 from the baseline. On the AES metric, 70% of reports have an equal or lower AES with X-REM compared to the baseline. X-REM reduces the average AES to 1.78 from the baseline AES of 2.48.

| | MES | | | | AES | | | |
|---|---|---|---|---|---|---|---|---|
| | 0 | $\leq 1$ | $\leq 2$ | $\leq 3$ | 0 | $\leq 1$ | $\leq 2$ | $\leq 3$ |
| X-REM (N=118) | 0.18 | 0.36 | 0.48 | 0.87 | 0.24 | 0.47 | 0.68 | 0.91 |
| Baseline (N=69) | 0.09 | 0.32 | 0.45 | 0.86 | 0.10 | 0.33 | 0.51 | 0.84 |
| Human Benchmark (N=53) | 0.34 | 0.49 | 0.64 | 0.94 | 0.35 | 0.56 | 0.69 | 0.94 |

Table 2: Human Evaluation Study Results. The table shows the empirical cumulative distribution for each report source on MES and AES scoring. For example, 87% of reports generated by X-REM received max error severity of 3 or less.

| | Metric | Filter | i | k | PT | RadCliQ $\downarrow$ | RadGraph $F_1$ $\uparrow$ | CheXbert $\uparrow$ | BERTScore $\uparrow$ | BLEU2 $\uparrow$ |
|---|---|---|---|---|---|---|---|---|---|---|
| 1 | ITM | NLI | 50 | 2 | ✓ | 3.781 | 0.133 | 0.384 | 0.287 | 0.084 |
| 2 | ITM | BS | 50 | 2 | ✓ | 3.801 | 0.130 | 0.387 | 0.281 | 0.088 |
| 3 | CS | N/A | 50 | 2 | ✓ | 4.017 | 0.122 | 0.336 | 0.244 | 0.081 |
| 4 | ITM | N/A | 50 | 2 | ✓ | 3.790 | 0.134 | 0.391 | 0.279 | 0.086 |
| 5 | ITM | NLI | $10^3$ | 2 | ✓ | 3.788 | 0.131 | 0.383 | 0.287 | 0.085 |
| 6 | ITM | NLI | 50 | 1 | ✓ | 3.767 | 0.109 | 0.420 | 0.279 | 0.072 |
| 7 | ITM | NLI | 50 | 3 | ✓ | 3.869 | 0.134 | 0.358 | 0.273 | 0.083 |
| 8 | ITM | NLI | 50 | 2 | | 4.387 | 0.072 | 0.250 | 0.195 | 0.055 |

Table 3: Ablation of models generating impression with image-text matching score (ITM) and cosine similarity score (CS) for the final filter (row 1,2), natural language inference (NLI) and BERtScore (BS) for the similarity metric at the retrieval step (row 3,4), parameter values for $i$ and $k$ (row 5,6,7), and pre-training (PT) the model (row 8). Row 1 reports the original version of X-REM.

**Ablation** Row 1, 2, and 4 of Table 3 compare the effect of appending a post-processing filter under the following three settings: appending an NLI-based filter (row 1), appending a BERTScore-based filter (row 2), and not appending a filter (row 4). For the BERTScore-based filter, we select the candidate text $r$ when $\text{BERTScore}_{F_1}(B, r) < 0.5$. We observe that the NLI-based filter gives the best RadCliQ score by optimizing on the BERTScore.

Row 3 and 4 of Table 3 compare the performance of cosine similarity and image-text matching score as the final similarity metric used in the retrieval. When we compare the retrieved reports (without a filter at the end), we observe that image-text matching score outperforms cosine similarity across every metric. Moreover, the vanilla image-text retrieval module with cosine similarity (row 3) still outperforms CXR-RePaiR from Table 1, which can be attributed to the performance gap of the base language-image models.

Row 5, 6, and 7 of Table 3 evaluate the model performance under various values for $i$ and $k$. The small gap between row 1 and 5 suggests that cosine similarity is useful for pruning irrelevant reports in the earlier stages. While technically X-REM with $k = 1$ (row 6) earns the best RadCliQ score, note that X-REM with $k = 2$ (row 1) receives a noticeably higher score than $k = 1$ across every metric except for CheXbert, which is why we use $k = 2$ in the final model.

In row 8, there is a significant drop in the performance of a model that was only fine-tuned from the published pre-trained checkpoint compared to row 1, which has been both pre-trained and fine-tuned on MIMIC-CXR. The difference suggests that radiology-related images and texts are vastly different from the data ALBEF was initially pre-trained on (Conceptual Captions (Sharma et al., 2018), SBU Captions (Ordonez et al., 2011), COCO

(Lin et al., 2014), and Visual Genome (Krishna et al., 2016)) and that additionally pre-training the language-image model on domain-relevant data is necessary.

## 5. Conclusion

We propose X-REM, a retrieval-based radiology report generation method that uses image-text matching score for report retrieval. Our quantitative analysis and human evaluation show that X-REM outperforms multiple prior radiology report generation modules. We attribute this increase in performance to X-REM's multi-modal encoder that uses a learned similarity metric to better capture the interaction between the X-ray images and radiology reports.

However, there still exists a notable gap between reports written by radiologists and generated by AI models including X-REM. The purpose of the human evaluation was to measure the size of the gap, and we plan to release our dataset so that other research efforts can contribute. Given the radiology report generation model's potential application in a medical setting, the models should be evaluated with a high standard. While our model has been optimized on clinical and natural language metrics, there may exist other important factors to reflect during the evaluation process. We focused on training and testing models on a single medical dataset that may contain bias, and the model's out-of-domain performance has not been empirically shown yet.

## 6. Acknowledgement

This project was supported by AWS Promotional Credit.

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

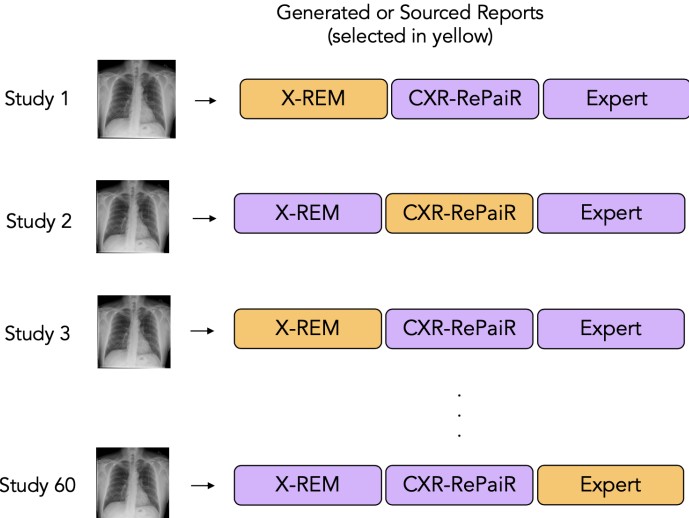

Figure 4: Human study randomization process. For each study, we chose a report from the three sources and presented the pair of chest X-ray and radiology report to human radiologists for evaluation.

