# OpenReview forum: "Multimodal Image-Text Matching Improves Retrieval-based Chest X-Ray Report Generation"
_MIDL.io/2023/Conference — MIDL 2023 Poster_

### Official Review · Reviewer_N8Wq · 2023-01-29

**Confidence:** 3
**Preliminary Rating:** 4
**Recommendation:** Poster

**Summary:**

This paper proposes a retrieval-based radiology report generation method. They extend the state-of-the-art by jointly representing image and text in a multi-modal language model, using an image-text matching score as a main similarity measure in the retrieval step. Additionally, the use supervised contrastive learning to fine-tune the model.
The method is evaluated on a public dataset of chest radiographs and radiology reports (MIMIC-CXR) and compared to several baselines. An ablation and a human evaluation study are conducted, showing the improved performance of the model


**Strengths:**

- Evaluation on benchmark data, which allows fair comparison to previous works. Additionally, the authors designed and conducted a qualitative evaluation by human experts.
- The ablation study is well designed and conducted, showing the effects of different parameters and modules.
- A section is dedicated to the limitations of the method.
- Good introduction, motivation and related work.


**Weaknesses:**

- The contributions/main differences to the state-of-the-art method by Endo et al., 2021 are not clear enough.
- The method section would need some revision. A clear high-level overview is missing. Fig. 1 is only to some extend helpful.
- A discussion/conclusion section is missing.
- A representative example of image and reports should be shown.


**Deanonymize Review:**

no

**Detailed Comments:**

- The difference to Endo et al., 2021 should be emphasized. In my understanding, the differences are that the encoder is multi-modal and that the retrieval is based on image-text matching score instead of cosine similarity. But Endo et al., 2021 also perform contrastive learning. Where are the differences here? If there are any, they have not been evaluated.
Is therefore the method from row 3 of the ablation study in principle the same as Endo et al., 2021 but with multi-modal encoders?
- The method section is not well structured and needs some revision, e.g.
Sec. 3.1: “As image-text matching score has been fine- tuned to learn the multimodal interaction, it can inherently make a more sophisticated judgment than cosine similarity which lacks such additional tuning. “
Fine-tuned on what? What is the model? How is the fine-tuning performed? How is the multi-modal interaction realized? This information is missing so far.
Fig. 1 is supposed to give an illustrative overview, but I found the figure very confusing. Only in the first paragraph of Sec 3 refers to the figure, but only the lower part (inference). Please also add more details in the caption.
- Some more explanations on the evaluation metrics is necessary (BLEU2, BERTScore). Also, please indicate in the tables if the metrics are to be minimized or maximized for best performance (similar to RadCliQ in Table 3).
- Human evaluation study: Please explain CDF, MES, AES in the table’s caption. Why do they have a different number of reports for the methods (N=117, N=69, N=53)? In the description of the experiment, they say that they use 60 studies. But then they only compare 40 paired studies for each method. An explanation is necessary.
- In the ablation study, the best RadCliQ value is achieved without filtering. So why is filtering necessary?
- Table 3: Explanation of abbreviations in the table and caption are necessary (P-T, ITM)
- There is more explanation/information necessary on the upper bound performance for retrieval-based methods. What is the upper bound and how close is the proposed method to it?
- “ALBEF directly aligns the embeddings of its image and text encoders, allowing us to its encoders both jointly to compute the image-text matching scores and separately to compute the cosine similarity. ”
Please rephrase. I don’t understand the sentence. Some typos?


**Paper Type:**

methodological development

**Questions To Address In The Rebuttal:**

- What are the main contributions of this work compared to Endo et al., 2021? Please clearly emphasize the main modifications
- How does the multi-modal encoders affect the results? If this is one contribution, it needs to be evaluated properly.
- Please revise section 3. A more detailed overview of all the different parts of the method, training etc would be extremely helpful.

---

### Official Review · Reviewer_CA95 · 2023-02-07

**Confidence:** 4
**Preliminary Rating:** 4
**Recommendation:** Poster

**Summary:**

This paper proposes a matching-based Chest X-ray report generation method. The authors use the language-image model to extract the features from both modalities. The language-image model is fine-tuned using paired data. The matching score that is calculated from image and text embedding is used as a similarity metric. The method is validated on MIMIC-CXR.

**Strengths:**

1. The paper is clear-written and easy to follow.
2. Several baseline methods are compared.
3. User-study and sufficient ablation studies were conducted.
4. Limitations are discussed and the authors provide insights in the evaluation process.

**Weaknesses:**

1. Although the authors illustrate the two key differences from the existing work, it is not clear how these differences can benefit the tasks. It is also not clear why the existing retrieval-based methods cannot achieve satisfactory performance.
2. Figure 1 is confusing as the modules in the inference stage are not well-aligned with those in the pertaining and fine-tuning stage. It will be better if the authors can link the encoder and decoder to the previous two stages.
3. Only one baseline method was used in the user study without justification for the selection.
4. Still, there exists a significant performance gap between the proposed method and the experts' performance.
5. From the ablation study, I may still miss the point of why ITM was used rather than cosine.
6. The training requires paired data. However, it is more common that the image and text data are unpaired in medical settings. I wonder if the method can be generalized to unpaired settings.

**Deanonymize Review:**

no

**Paper Type:**

methodological development

**Questions To Address In The Rebuttal:**

Could you please address the points listed in the weakness?
In addition, could you please highlight the novelty of the proposed method vs the alternative methods from more technical views? Could you please elaborate more on the Top K corpus concatenation step and the natural language inference filter step?

---

### Official Review · Reviewer_ksHb · 2023-02-07

**Confidence:** 3
**Preliminary Rating:** 4
**Recommendation:** Oral, Poster

**Summary:**

This paper presents a novel retrieval-based radiology report generation technique where they adopt multiple filters within a CLIP pipeline. A cosine filter acts as the primary barrier after measuring the similarity between the embeddings of a Chest X-ray and the existing retrieval corpus and taking the top $i$ reports. Following that, both the top $i$ reports and the Chest X-ray scan is fed into a multi-modal encoder, to take the top $j$ reports based on the classification-based match between the reports and the scan. Finally, the reports are transferred into a natural language inference engine to omit the contradicting and entailing reports.

**Strengths:**

1. Novel topic
2. Includes an ablation study to measure the effect of image-text matching metric, filtering methodology, top-I, top-k, and pre-training.
3. Achieves the best performance in 4/5 metrics comparing a variety of baselines
4. An additional reader’s study to assess the quality of the compiled reports


**Weaknesses:**

1. The paper needs some improvement in terms of the use of language and proper referencing
2. Needed to check prior work to understand the baseline methods and evaluation metrics
3. Conclusion & Discussion section is not present
4. No source code has been made available


**Deanonymize Review:**

yes

**Detailed Comments:**

* Can you provide some visual demonstrations in which you provide chest X-ray scans for the input, the generated reports, and the expert evaluation scores for each line within generated reports?
* Can you provide what CDF and PT stand for within the description of the tables? Please also check the rest of the paper accordingly.
* Is the natural language inference filter prone to repeatability, e.g., would the generated report change after running the framework with the same input at once more?
* Can’t see your conclusions?
* RadGraph F_1 has been developed in (Yu et al., 2022), not in (Jain et al., 2021).
* To perform the human benchmark in Table 2, I notice that you take a ground truth report from the MIMIC-CXR dataset and ask the radiologists to mark each line, right? If that is correct, can we say that there are some urgent and emergent errors present in the reporting part of the MIMIC-CXR dataset? Or can we say that it is because of interrater variability? What exactly do the numbers within the parentheses mean, considering that there are 60 studies in total?
* Is there a logical error in the paired comparison section, e.g., P(X-REM > baseline) = 55%, P(X-REM <= baseline) = 62.5% ?
* Try to use some boxes in Figure 1 to demonstrate the procedures more clearly.
* Do you intend to publish the source code of this paper?


**Paper Type:**

methodological development

**Questions To Address In The Rebuttal:**

Please try to address the questions mentioned above during the rebuttal period. Specifically, I am eager to see a well-formed text in addition to the answers to my questions about methodology and experiments.

---

### Meta-Review · Area_Chair_LXFc · 2023-02-22

**Recommendation:** Accept (Poster)
**Confidence:** 4

**Metareview:**

The paper presents a novel method for automatic radiology report generation. The reviewers agree on the novelty of the topic, the strength of the ablation study and limitations section. Reviewers raise various questions about novelty and implementation of the method, which in my view the authors have mostly been able to address in the responses and revised PDF. Overall, I recommend acceptance of the paper.

P.S. Regarding code release for anonymized papers, it is possible to use https://anonymous.4open.science/